# Single-Shot Scene Reconstruction

**Sergey Zakharov**[1], **Rareş Ambruş**[1], **Vitor Guizilini**[1], **Dennis Park**[1], **Wadim Kehl**[2],
**Fredo Durand**[3], **Joshua B. Tenenbaum**[3], **Vincent Sitzmann**[3], **Jiajun Wu**[4], **Adrien Gaidon**[1]

[1]Toyota Research Institute, [2]Woven Planet, [3]Massachusetts Institute of Technology, [4]Stanford University

**Abstract:** We introduce a novel scene reconstruction method to infer a fully editable and re-renderable model of a 3D road scene from a single image. We represent movable objects separately from the immovable background, and recover a full 3D model of each distinct object as well as their spatial relations in the scene. We leverage transformer-based detectors and neural implicit 3D representations and we build a Scene Decomposition Network (SDN) that reconstructs the scene in 3D. Furthermore, we show that this reconstruction can be used in an analysis-by-synthesis setting via differentiable rendering. Trained only on simulated road scenes, our method generalizes well to real data in the same class without any adaptation thanks to its strong inductive priors. Experiments on two synthetic-real dataset pairs (PD-DDAD and VKITTI-KITTI) show that our method can robustly recover scene geometry and appearance, as well as reconstruct and re-render the scene from novel viewpoints.

## 1 Introduction

Decomposing images into disjoint symbolic representations is the ultimate goal of robotics and computer vision, as it allows semantic reasoning over all scene parts. Such a decomposition benefits different possible applications (e.g., robotics, augmented reality, autonomous driving) where a decomposed scene can be reassembled in different ways, enabling interaction or reenactment. Unfortunately, single image scene reconstruction is inherently ill-posed due to a variety of reasons. Most prominently, ambiguity in projective geometry can only be resolved with prior knowledge about the observed scene. Current state-of-the-art methods either do not provide a full decomposition of the scene and regress per-pixel depth [1, 2], or estimate the poses of already known objects and ignore the rest of the geometry [3, 4].

In this work, we design a pipeline to obtain an interpretable, disentangled scene representation for 3D-aware manipulation of driving scenes. Our system is capable of reconstructing full geometries and appearances of detected object instances from a single RGB image. Such a decomposition is made possible by three components: (1) a scene decomposition network (SDN) that detects object instances, recovers their partial geometries and poses, and predicts the full geometry of the background and RGB color behind the occluders; (2) a differentiable database of colored object priors that encodes their full geometry and appearance in the form of signed distance fields (SDF) and luminance fields (LF); and (3) a 2D/3D optimization pipeline that uses surfel-based differentiable rendering to fit initial observations to the object database, leveraging their shape and appearance.

Our contributions are summarized as follows:

- A holistic single-shot scene reconstruction system that splits the scene into background and object instances and recovers their full geometry and appearance;
- An optimization pipeline based on NOCS-SDF and RGB-LF fitting as well as surfel-based differentiable rendering to refine geometries, appearances, and poses of detected instances;
- A scene manipulation and generation pipeline allowing to place and render objects of various geometry and appearance, and change view points.

5th Conference on Robot Learning (CoRL 2021), London, UK.

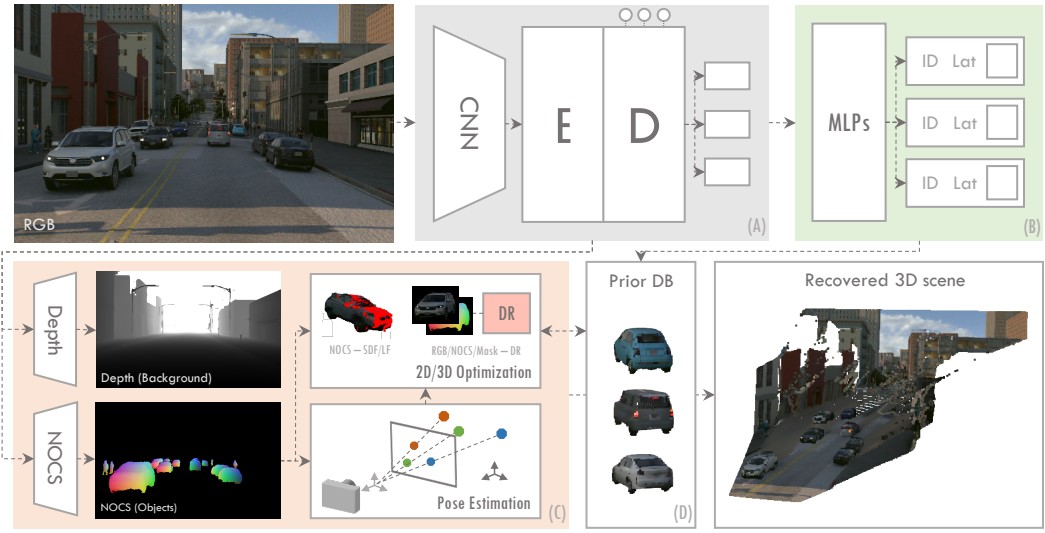

Figure 1: **Scene Decomposition Network (SDN)**: Given an RGB image, our SDN recovers partial canonicalized object shapes in the form of NOCS maps and predicts background depth. NOCS maps are then used to directly estimate object poses using a P*n*P solver. We further fit learned object priors to align with predicted partial shapes in terms of geometry, appearance, and poses.

## 2 Related Work

**Single-Shot Scene Reconstruction**  Holistic single image scene reconstruction is a highly ill-posed problem, tackled from different angles by a variety of works. The most common approach is monocular depth estimation that predicts per-pixel depth values [5, 1, 2]. While achieving re-markable results, these methods reconstruct the scene as a whole and do not incorporate knowledge about the scene's objects and layout. Alternatively, 3D detection pipelines detect separate objects and recover their masks and 3D bounding boxes [3, 6, 7, 4], or incorporate relationships between objects by using a graph or physical simulation [8, 9, 10]. While having understanding about the object appearance in 2D and their 3D pose, these methods do not recover the shapes of the objects nor the rest of the scene. There are also pipelines aiming at reconstructing objects from detections by aligning 3D CAD models to fit the predicted masks and input images [11, 12, 13], albeit without generalization to unknown instances outside the model database. Lastly, there are recent works that recover room layouts together with full object geometries [14, 15], but restrict themselves to the geometric components of the recovered instances. In our work we recover the geometry and appearance for both background and detected objects by using a differentiable database of object priors allowing us to smoothly interpolate between plausible object shapes.

**6DoF Object Detection via Correspondence Regression**  Object detection and pose estimation from RGB images is a well-established but ongoing research problem due to, among others, the inherent ambiguity of perspective projection. The myriad of existing solutions can loosely be grouped in three categories: direct pose regression (using a neural net to output the pose directly by learning from data) [16, 17, 18, 19], template-matching (comparing patches with a predefined set of templates either in pixel space or feature space) [20, 21, 22, 23, 24, 25], and correspondence-based methods (directly utilizing correspondences between image coordinates and 3D models of interest) [26, 27, 3, 4] - each having their own benefits and drawbacks. The current state-of-the-art methods in object pose estimation almost exclusively belong to the latter group with such representatives as PVNet [7], CDPN [28], EPOS [27], Pix2Pose [4], GDR-Net [29] and DPOD [3, 6]. They all regress per-pixel correspondences and use a P*n*P solver to estimate a 6DoF pose given a set of 2D-3D correspondences and camera parameters, minimizing reprojection errors. While some works regress manually defined UV correspondences, others regress *Normalized Object Coordinate Space* (NOCS) maps [30] that represent visible $xyz$ object coordinates in RGB space. The benefits of NOCS are their straightforward creation through rendering, and their simultaneous definition of both the object's pose and shape. In our work we show how to effectively leverage this representation for geometry retrieval as well.

**Neural Implicit Representations**   Recently, a new prominent direction for 3D shape and appearance representation has emerged with DeepSDF [31], Occupancy networks [32], and IM-Net [33], using neural networks for scalar function approximation. All three methods take 3D coordinates as input and output either a binary occupancy estimate or a continuous SDF value, encoding the object's surface. Subsequently, follow-up work such as Deep Local Shapes (DLS) [34], Neural Geometric Level of Detail (NGLOD) [35] or NASA [36] improved the direction further. While DLS stores SDFs as voxel grid cells to allow larger scenes, NGLOD uses an efficient Octgrid representation and a feature composition scheme based on trilinear interpolation to further improve the reconstruction quality, whereas NASA extends the approach to articulated objects.

Similar representations have been extended to represent object appearance. Scene Representation Networks (SRN) [37] add a ray marching routine and also regress RGB colors at surface intersections to learn from multi-view images, and Differentiable Volumetric Rendering [38] which couples an implicit shape representation with differentiable rendering. Alternatively, instead of regressing SDF or RGB values for predefined 3D coordinates, NeRF [39] proposes to regress density and color values along rays (5D coordinates) and compute the true color via numerical integration. This approach, drawing inspiration from Neural Volumes [40] and their volumetric representation, boosted interest in implicit volumetric rendering and resulted in a multitude of works tackling problems from training and rendering time performance [41, 42, 43, 44], to covering dynamic scenes [45, 46, 47, 48], scene relighting [49, 50, 51], and composition [52, 53, 54].

While providing realistic renderings, most NeRF-based methods overfit to a single scene (with the exception of PixelNeRF [55]) and require a set of very dense view point annotations. Our pipeline, on the other hand, is capable of reconstructing scenes never seen during training, and is also not limited to viewpoints seen during training.

## 3   Methodology

Given a single RGB image of a typical driving scene, our pipeline holistically reconstructs the scene by not only explaining every visible pixel, but also predicting the full geometry and appearance of detected objects and the occluded areas that are not visible in the image. This is made possible by a combination of three components: a scene decomposition network (SDN), a differentiable database of object priors, and a 2D/3D optimization pipeline.

### 3.1   Scene Decomposition Network

The first component of our pipeline (Fig. 1) is the scene decomposition network, or SDN. Its main role is to estimate the layout of the scene, detect and identify visible objects, and recover the geometry of both the background and the detected objects. For clarity, it is divided into three separate blocks: a detection transformer-based decomposition block, an object reasoning block, and a 3D reasoning block.

**Detection Transformer Block (Fig. 1a)**   The core of our SDN is a transformer-based decomposition. An encoder-decoder object detector based on transformers, a popular architecture for sequence prediction and has been first applied to computer vision in DETR [56]. DETR first generates distinctive image features from an input RGB image, and feeds them to a transformer module to build attention maps between features. This self-attention mechanism explicitly models all pairwise interactions between elements in a sequence, making this architecture particularly suitable for specific constraints of set prediction (e.g. duplicate suppression). Using the notion of object queries, we are able to retrieve output features for each of the detected objects, which are then used as input to our object reasoning block and 3D reasoning block.

**Object Reasoning Block (Fig. 1b)**   Our object reasoning block takes output features of the detection transformer and uses a collection of MLPs to predict important object properties. Following [56] we use a 3-layer perceptron with ReLU activation and a linear projection layer to regress object class IDs and 2D bounding boxes. Additionally, we also regress SDF and LF feature vectors, denoted by $\mathbf{z}_{sdf}$ and $\mathbf{z}_{lf}$ respectively. These feature vectors provide an initial state for object reconstruction and are essential to effectively reason about and refine the 3D component of these objects, as described in detail in Section 3.2.

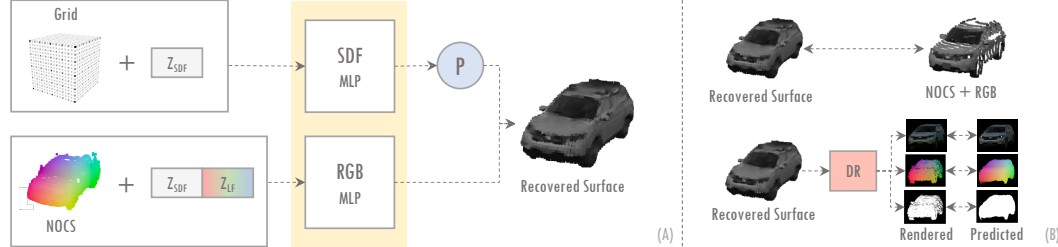

Figure 2: **Differentiable database of object priors (A)** and **shape, appearance, and pose optimization (B)**. Our object priors are learned using coordinate-based MLPs, taking positionally encoded features as input and recovering geometry in form of SDF and appearance on the object's surface. We use a 0-isosurface projection (marked P), to retrieve object's surface from an SDF field. Our initial recovered object is then refined to fit both predicted observations in terms of masks and NOCS map, as well as ground truth RGB values from the input image.

**3D Reasoning Block (Fig. 1c)**   Our 3D reasoning block aims to recover 3D scene information by splitting it into two parts: a background containing road surfaces, buildings, and other objects not detected by the detection transformer, and a foreground entirely consisting of detected objects. We use the output of the transformer decoder for each object to compute multi-head attention scores of this embedding over the output of the encoder, generating M attention heatmaps per object. These masks are then used to regress the geometry for both foreground and background.

The foreground is predicted as a set of normalized shape maps. Since they encode 3D coordinates via RGB, visualizing each 3D color component in 3D space allows us to recover a partial 3D shape of the object in its normalized coordinate space (Fig. 3). This representation is crucial for our pipeline to retrieve the object's full geometry and pose.

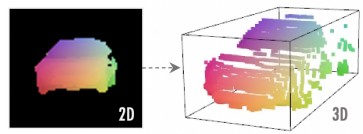

The background, on the other hand, is represented as a depth map since we solely care about its geometry. For view synthesis applications, we also predict the depth and appearance

Figure 3: Partial shape recovery from a predicted 2D NOCS map.

behind the detected objects by utilizing a GAN-like encoder-decoder generator architecture. This architecture takes masked RGB and depth images and inpaints RGB and depth values for occluded regions, employing a fully-convolutional PatchGAN discriminator to judge the genuineness of the inpainted maps.

Given estimated object masks and NOCS maps, we recover three things enabling pose estimation: (1) 2D coordinates of the object in the image, (2) 3D points of the partial object shape in the canonical frame of reference, (3) and their 2D-3D correspondences. This unique representation frees us from the need of storing a collection of 3D primitives and identifying them to find a detected model, because both 3D and 2D information is contained in the form of a 3-channel map. We multiply the recovered normalized shape by the per class scale factor to recover absolute scale. The 6DoF pose is then estimated using P$n$P [57] that predicts the object pose from given correspondences and camera intrinsics. Since we get a large set of correspondences for each model, we also combine P$n$P with RANSAC to robustify against outliers. For the results presented in the evaluation section, we run 1000 RANSAC iterations per pose with a $1px$ reprojection error threshold.

## 3.2   Shape, Pose, and Appearance Optimization

The second crucial component of our pipeline is a 2D/3D optimization that is enabled by the differentiable database of object priors (Fig. 2a) as well as our surfel-based differentiable rendering pipeline (Fig 2b). Given output features corresponding to detected objects regressed by the object reasoning block (Fig. 1b), our optimization procedure generates full shapes, appearances and poses of all the objects in the scene.

We leverage a differentiable database of object priors - *PriorDB* - which encodes the shape and luminance of the input models as Signed Distance Fields (SDFs) and Luminance Fields (LFs) with associated latent feature vectors. Given a partial shape observation, we differentiate against *PriorDB*

and find the maximum likelihood latent vector that best explains the observed shape. The RGB component is optimized similarly, with the complete procedure described in detail in Sec 3.2.1.

While NOCS-based SDF and LF optimization allows us to reconstruct plausible objects from images, it does not account for the pose component. It might very well be the case that estimated poses are incorrect and are harming the full scene reconstruction. This can be alleviated via a 2D alignment step, which can further be used as a complimentary source for shape optimization when predicted NOCS maps are noisy, resulting in a lower reprojection error and better reconstructions. Thus, to further constrain and improve the optimization, we define a differentiable rendering pipeline that allows us to impose 2D image losses based on renderings of the recovered SDF shape and appearance, as described in detail in Sec. 3.2.2.

### 3.2.1 Differentiable Database of Object Priors (Fig. 1d)

Our differential database of object priors or *PriorDB* represents objects as signed distance fields (with positive and negative values pointing to exterior and interior areas respectively), in which each value corresponds to the distance to the closest surface. A single MLP can be used to represent multiple geometries using a single feature vector $\mathbf{z}_{sdf}$ and query 3D locations $\mathbf{x} = \{x_1, ..., x_N\}$ as $f_{sdf}(\mathbf{x}; \mathbf{z}_{sdf}) = \mathbf{s}$. Object appearances, on the other hand, are represented as luminance fields (LF), defining the prior perceived luminance of the object as seen in the training set. Similarly to SDN, the LF module is implemented as an MLP, but it takes a feature vector by concatenating $\mathbf{z}_{sdf}$ and $\mathbf{z}_{lf}$ as well as query locations $\mathbf{x}$ as input, and outputs resulting luminance as $f_{lf}(\mathbf{x}; \mathbf{z}_{sdf}; \mathbf{z}_{lf}) = \mathbf{l}$.

Both SDF and LF modules are trained using synthetic data provided by Parallel Domain [58], containing high-quality realistic road scene renderings together with ground truth annotations. To train the SDF module, we use a collection of canonical ground truth point clouds and associated surface normals representing rendered objects. In total, we encode 27 cars in our *PriorDB*. We first define a multi-level octree spanning the volume $[-1, 1]^3$, with points located at the corners of the imaginary voxels of various sizes depending on the level. For each of

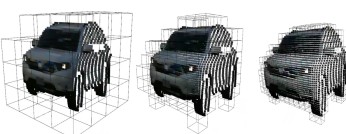

Figure 4: Different octree levels used for training SDF primitives.

the point clouds we define unsigned distances by computing nearest neighbors to the octree points. Next, we define the loss $L_{sdf}$ as the distance to the surface at grid points starting from the coarsest level, gradually increasing the resolution up to the highest level (we use 3 levels of granularity as shown in Fig. 4), while only considering points located at pre-defined per-level distances to the surface. To ensure that we learn a correct signed field, we additionally minimize the distance between the ground truth surface normals and analytically-derived surface normals from the SDF module. Moreover, to ensure that we always regress plausible shapes regardless the input feature vector, we do an additional refinement round by sampling random feature vectors and enforcing the output to be equal to the shape of the nearest plausible neighbor.

Once the SDF module is trained, we store SDF features $\mathbf{z}_{sdf}$ associated with objects and use them to train the LF module. Differently, it is trained on partial canonical shapes recovered from provided RGB renderings (see Fig. 2a), associated depth maps and poses. We compute the luminance component from RGB renderings and enforce the LF of each feature to be the same. The network weights are then optimized to fit the average luminance per object using an L1 loss. Additionally, we define a symmetry loss $L_{symm}$ enforcing colors on the left side of the car to be similar to those on the right one. For that we compute nearest neighbors between the left side of the car and an inverted right side and then minimize the L1 distance between these correspondences.

**Optimization 3D** Our optimizer takes partial canonical object shape and appearance observations (recovered from predicted NOCS maps as shown in Fig. 1b), as well as initial *PriorDB* features. Due to the robustness of NOCS regression, and despite the occasional noisy outliers, we set it as our main optimization prior. The initially predicted SDF is used to recover the full surface of the object in the form of a point cloud with surface normals using a 0-isosurface projection. Then we estimate nearest neighbors between the two point clouds and minimize the distance between the points by optimizing the feature vector $\mathbf{z}_{sdf}$. The RGB component is optimized similarly by minimizing the color difference between the correspondences, but we also allow the LF module weights to vary for a finer level of detail. In the case of RGB optimization we also use the $L_{symm}$ loss between the left and right sides of the predicted shape.

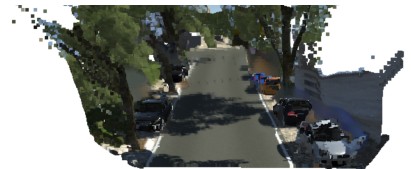 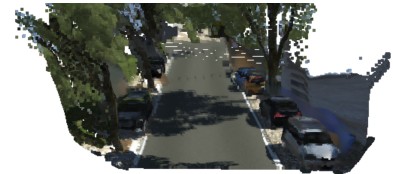

Figure 5: **VKITTI2 reconstructions.** Our SDN recovers partial reconstructions (left), which are then completed with shape priors from *PriorDB* using our 2D-3D optimization pipeline (right).

### 3.2.2 Surfel-based Differentiable Renderer

Differentiable rendering allows to optimize objects with respect to pose and shape, and is an essential component of our optimization scheme. Since we use an SDF representation, we cannot directly use differentiable renderers for triangulated meshes. Instead, we implemented a renderer that uses surfels as the primary representation (inspired by [59]), estimating surface points and normals using an 0-isosurface projection, followed by a surfel-based rasterizer.

**Surface Tangent Discs** Surface elements or surfels are a common concept in computer graphics [60] as an alternative to connected triangular primitives. In order to render watertight surfaces, the individual surface normals must sufficiently approximate the local region of the object's geometry.

$$d = \frac{n_i \cdot p_i}{n_i \cdot \mathbf{K}^{-1}(u, v, 1)^T} \quad \text{(1a)} \qquad P = \mathbf{K}^{-1} \cdot (u \cdot d, v \cdot d, d)^T \quad \text{(1b)}$$

To construct surface discs we first estimate the 3D coordinates of the resulting tangent plane given the normal $n_i = \frac{\partial f(p_i; \mathbf{z})}{\partial p_i}$ of a projected point $p_i$. The distance $d$ of the plane to each 2D pixel $(u, v)$ is computed by solving a system of linear equations for the plane and camera projection as defined in Eq. 1a, where $\mathbf{K}^{-1}$ is the inverse camera matrix. Then, we get a get a 3D plane coordinate by backprojection (Eq. 1b). Finally, we estimate the distance between the plane vertex and surface point and clamp if it is larger than a disc diameter to get final discs $M = \max(diam - ||p_i - P||_2, 0)$.

**Rendering Function** Similarly to [59, 61], we combine colors from different surfel primitives based on their depth values to compose a final rendering. The distance of the primitive to the camera defines its final intensity contribution to the image. To ensure that all primitive contributions sum up to 1 at each pixel we use a modified softmax function. The final rendering function is given in Eq. 2a, where $\mathcal{I}$ is the output image, $S$ the estimated NOCS map, and $w_i$ the weighting masks,

$$\mathcal{I} = \sum_i S(p_i) * w_i, \quad \text{(2a)} \qquad w_i = \frac{exp(-\tilde{D}_i \boldsymbol{\sigma}) M_i}{\sum_j exp(-\tilde{D}_j \boldsymbol{\sigma}) M_j}. \quad \text{(2b)}$$

Eq. 2b defines weighting masks $w_i$, where $\tilde{D}$ is the normalized depth and $\boldsymbol{\sigma}$ is a transparency weight with $\boldsymbol{\sigma} \to \infty$ defining a completely opaque rendering as only the closest primitive is rendered.

**Optimization 2D** Formally, we define three losses on the rendering output $L_{nocs_{2D}}$, $L_{lf_{2D}}$, and $L_{mask_{2D}}$. $L_{nocs_{2D}}$ compares the output of the renderer with predicted NOCS, $L_{lf_{2D}}$ compares the output of the renderer with input RGB images, and $L_{mask_{2D}}$ compares the rendered and the predicted masks. The exact definitions of these losses are provided in the appendix. The final optimization loss is a weighted sum of both 2D and 3D components (see Sec. 3.2.1). While 3D components ensure a robust shape optimization, 2D losses help to better align the object in 3D space and allow a better luminance matching.

## 4 Experiments

In this section, we first evaluate the individual components of the SDN on common benchmarks for stereo view synthesis, depth estimation and 3D object detection. Then we demonstrate the capabilities of our full pipeline for the tasks of scene reconstruction, object recovery and manipulation. The evaluation is performed on four datasets: PD, VKITTI2, DDAD, and KITTI.

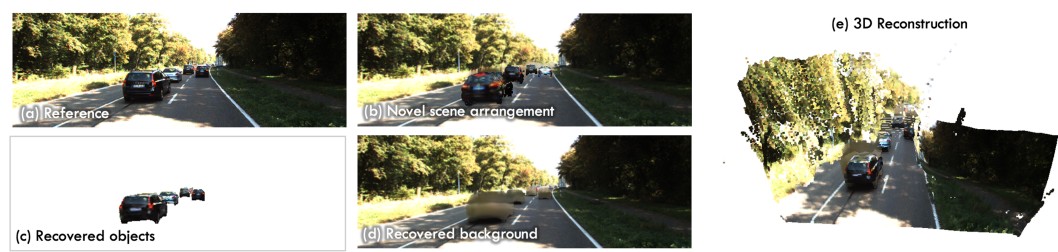

Figure 6: **Scene reconstruction using our pipeline on KITTI.** Our pipeline decomposes the scene into objects (c) and background (d) and recovers full geometries and appearances (e). This allows for scene manipulation, view synthesis and novel scene arrangements (b). Despite being trained on synthetic data, our pipeline generalizes well to real images without any adaptation.

**VKITTI2 [62] and KITTI [63]**    Virtual KITTI [64] is a synthetic proxy to the KITTI dataset – the standard benchmark for autonomous computer vision. It contains five virtual worlds rendered under ten different conditions of weather and camera configuration, in total comprising 21,260 images. VKITTI2 extends the original dataset with more photo-realistic renderings. In the experiments we use the first 80% consecutive frames of each configuration for training and the last 20% of the *clone* configuration for testing. To evaluate object detection and depth estimation on real data we use the KITTI-3D [65] validation split consisting of 3769 images.

**Parallel Domain (PD) [58] and DDAD [2]**    The Parallel Domain dataset is a procedurally-generated collection of fully annotated photo-realistic renderings of urban driving scenes. It contains 200 180-frame sequences resulting in a total of 36000 images. Each sequence also contains an additional frame with extra ground truth modalities which we use to form the PD *test* split with a total of 200 images. The Dense Depth for Automated Driving (DDAD) dataset is a real-world counterpart of the PD dataset mainly used as a depth evaluation benchmark. We consider only the front camera, resulting in 50 validation sequences with 3950 images.

### 4.1   Scene Decomposition and 3D Reasoning

**Stereo View Synthesis**    In this experiment we quantitatively evaluate our SDN for the task of view synthesis by recovering a stereo-pair of the input image. We compare our results with state-of-the-art generalizable NeRF-based view synthesis method PixelNeRF [55] and depth estimation method PackNet-SfM [2]. All pipelines were trained on the same VKITTI2 *training* split and evaluated on the VKITTI2 *test* split. From the results in Table 2, we observe that SDN can reliably re-generate the scene from a novel view point thanks to its integrated prior and geometric nature, and outperforms both baselines. Please refer to the appendix for qualitative results.

**Depth Evaluation**    Next we evaluate the quality of our geometric output using common metrics for depth evaluation. We argue that our NOCS maps, given estimated poses, can be transformed

| Train | Method | Test | Lower is better | | | | Higher is better | | |
|---|---|---|---|---|---|---|---|---|---|
| | | | AbsRel | SqRel | RMSE | $RMSE_{log}$ | $\delta < 1.25$ | $\delta < 1.25^2$ | $\delta < 1.25^3$ |
| VKITTI2 | Monodepth2 | KITTI | 0.239 | 3.217 | 8.425 | 0.354 | 0.469 | 0.655 | 0.767 |
| | PackNet-SfM | | 0.211 | 2.746 | 7.701 | **0.319** | 0.546 | 0.704 | 0.780 |
| | **Ours** | | **0.175** | **2.375** | **7.384** | 0.360 | **0.642** | **0.756** | **0.803** |
| | Monodepth2 | DDAD | 0.437 | 15.621 | 26.578 | 0.731 | 0.025 | 0.118 | 0.241 |
| | PackNet-SfM | | 0.340 | 11.290 | 22.332 | 0.461 | 0.138 | 0.309 | 0.521 |
| | **Ours** | | **0.238** | **10.402** | **18.228** | **0.346** | **0.535** | **0.652** | **0.704** |
| PD | Monodepth2 | KITTI | 0.368 | 7.976 | 14.432 | 0.653 | 0.320 | 0.480 | 0.580 |
| | PackNet-SfM | | 0.332 | 3.779 | 9.205 | 0.395 | 0.318 | 0.672 | 0.836 |
| | **Ours** | | **0.176** | **2.407** | **7.687** | **0.372** | **0.696** | **0.795** | **0.840** |
| | Monodepth2 | DDAD | 0.187 | 3.951 | 6.465 | 0.184 | 0.429 | 0.506 | 0.586 |
| | PackNet-SfM | | 0.165 | **2.500** | **6.017** | **0.165** | 0.366 | **0.591** | **0.620** |
| | **Ours** | | **0.139** | 3.808 | 6.612 | 0.231 | **0.497** | 0.559 | 0.590 |

Table 1: **Depth estimation results** for direct transfer from synthetic (VKITTI2 and PD) to real (KITTI and DDAD) datasets. KITTI results are reported with the *Garg* crop at distances up to 80m, and DDAD results are reported without cropping and at distances up to 80m.

| Method | PSNR↑ | SSIM↑ | LPIPS↓ |
|---|---|---|---|
| PixelNeRF | 16.63 | 0.45 | 0.57 |
| PackNet-SfM | 17.88 | 0.64 | 0.36 |
| **Ours** | **18.51** | **0.66** | **0.31** |

Table 2: **View synthesis.** We evaluate the performance of our SDN for the task of stereo-pair recovery on the VKITTI2 dataset.

| Method | BEV AP | | | 3D AP | | |
|---|---|---|---|---|---|---|
| | Easy | Med | Hard | Easy | Med | Hard |
| SMOKE | 14.30 | 9.51 | 5.94 | **11.47** | 7.40 | 5.31 |
| **Ours** | **16.50** | **11.88** | **10.15** | 10.94 | **7.54** | **6.72** |

Table 3: **KITTI-3D detection.** Reported are $AP|_{R_{40}}$ metrics of *Car* class with 50% IoU threshold. Both methods are trained on VKITTI2 *train* set.

| | Shape (mm) | | Mask (IoU) | | RGB (PSNR) | |
|---|---|---|---|---|---|---|
| | Mean | Median | Mean | Median | Mean | Median |
| No opt. | 46.57 | 14.87 | 83.85 | 87.34 | 16.27 | 16.34 |
| 2D | 34.44 | 9.39 | 85.51 | 89.28 | 20.37 | 20.20 |
| 3D | **31.63** | 9.20 | 86.44 | 89.99 | 20.10 | 19.85 |
| 2D + 3D | 32.71 | **8.83** | **86.46** | **90.01** | **20.55** | **20.45** |

Table 4: **Optimization ablation.** Effect of optimization on object shape and luminance estimation, and 2D amodal mask alignment.

to depth maps and significantly outperform standard depth estimation pipelines due to the prior geometry knowledge. To conduct this experiment, we choose current state-of-the-art depth estimation pipelines, Monodepth2 [1] and PackNet-SfM [2], and train them on 2 synthetic datasets - VKITTI2 and PD - similarly to our method. Then we evaluate the depth quality for foreground objects on real datasets - KITTI and DDAD - using a number of standard depth estimation metrics. This experiment evaluates three aspects of our model: 1) the reconstructed object geometry, 2) the accuracy of the estimated poses, and 3) transferrability across domains. Table 1 shows that our model consistently outperforms the baselines for monocular depth estimation, demonstrating the superior geometric accuracy and generalization qualities achieved thanks to the encoded object prior.

**3D Detection** In this experiment we aim to demonstrate how well our method generalizes to other datasets in terms of 3D detection quality. We compare against a state-of-the-art method for KITTI pose estimation, SMOKE [66], which is trained on the same VKITTI2 *train* set as our method. To compare with our algorithm that has access to ground-truth size of 3D boxes, we substitute the size of predicted boxes with the ones of nearest-neighbor ground-truth, when evaluating SMOKE. Table 3 shows that we perform competitively with SMOKE, and demonstrates that our underlying NOCS representation generalizes to diverse shapes and poses of objects in the real-world.

## 4.2 Scene Manipulation and Object Recovery

The section aims to demonstrate the capability of our full pipeline to reconstruct and manipulate the scene. In Fig. 6, we show the scene decomposition results on the KITTI dataset. Our SDN takes as input a single reference image (Fig. 6a) and recovers full 3D objects (Fig. 6c) as well as the scene background (Fig. 6d). Recovered objects can be freely manipulated as shown in Fig. 6b. The full 3D scene reconstruction is shown in Fig. 6e. Note that our pipeline has been trained entirely on synthetic data, and generalizes to real images without any adaptation. We refer the readers to our supplementary videos for the visualization of the full capability of scene manipulation.

**Optimization ablation** Lastly, we investigate how different components of our optimization pipeline affect the end result in terms of full shape and luminance estimation, as well as 2D amodal mask alignment. All evaluations are performed on the PD *test* split. We observe a significant performance increase for all considered metrics when compared to an unoptimized result as shown in Table 4. While 3D optimization enables robust shape estimation, it does not align luminance in the projective space and also ignores object's pose since it is performed in the object's canonical frame. The 2D optimization step on the other hand is performed in the projective space and allows for exact luminance matching and good mask alignment. However, it performs worse with respect to shape optimization due to an additional ambiguity introduced by the pose component, which also affects the amodal mask metric. Finally, using both 2D and 3D optimization combines their strengths and yields the best overall performance.

## 5 Conclusion

We present a novel holistic scene reconstruction pipeline that disentangles the scene by representing objects separately from the scene. Our Scene Decomposition Network detects and recovers partial geometries and poses of objects, and estimates depth of the background. The partial geometries are then completed using our differentiable database of object priors and a 2D-3D optimization procedure. As a result, our pipeline explains every pixel on the image and more - the full object geometries and appearances. Despite being trained only on synthetic data our experiments show good generalization to real data on KITTI and DDAD datasets.

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
