# OpenReview forum: "Single-Shot Scene Reconstruction"
_robot-learning.org/CoRL/2021/Conference — CoRL2021 Poster_

### Official Review · Reviewer_jaEn · 2021-07-20

**Originality:** Good
**Technical Quality:** Good
**Clarity Of Presentation:** Fair
**Impact:** 4

**Recommendation:**

Weak Reject: I recommend rejecting the paper, but will not argue for my recommendation if the majority of other reviewers have a different opinion.

**Summary:**

This paper presented a new single view scene reconstruction method that can decompose the scene into the background and multiple objects (cars in the shown demo) and recover their full geometry and appearance. It also involves an optimisation step to refine the geometry, appearance and poses of the detected instances based on NOCS-SDF and RGB-LF fitting with a surfel-based differentiable rendering method. The quantitative evaluations show SOTA performance in terms of depth evaluation and 3D detection. It furthers shows that it can achieve scene manipulation and generation to enable novel view synthesis.

**Issues:**

In addition to some issues mentioned in the weakness part:
* L150: Fig.5a is not the figure referred does not include object prior database.

**Reviewer Expertise:**

Very good: Comprehensive knowledge of the area

**Strengths And Weaknesses:**

This paper presents a novel single-view holistic scene reconstruction/composition method. The strength is that it can recover the appearance by adding an extra latent embedding for LF. It also contains an optimisation capability using a surfel-based differentiable rendering approach, which makes the method applicable to unseen environments.

In terms of the weakness, there are a few points.
1. Comparison: In the related work, the authors perhaps missed some recent work in the single-view scene reconstruction that can also recover the full object geometry. Such as "Total3DUnderstanding: Joint Layout, Object Pose and Mesh Reconstruction for Indoor Scenes from a Single Image" (CVPR 2020), "Holistic 3D Scene Understanding from a Single Image with Implicit Representation" (CVPR 2021), and some others. The authors might need to add these work in the related work part and show how their work is positioned against those work. The mentioned works can also recover the room layout and generate multi-class instances. From my perspective, the strength of this presented paper is that it can also recover the full appearance of the detected instances. However, this point is not evaluated or demonstrated in the experiment part. This innovation needs a thorough evaluation.

2. Explanation of method: Section 3.2 needs a more clear explanation of the losses used for optimisation, for example, z_{sdf}, RGB loss, L_{symm}, L_{{NOCS_2D}}, L_{LF_{2D}}, L_{mask}, pose optimisation, normal distances. The equations for those losses and how they are summed under different circunsmaces are needed to be able to reproduce the results.

3. Evaluations: Some qualitative results are needed to demonstrate the numbers in the view synthesis. Besides, as the pose estimation is an important step in the pipeline, it is expected to see some evaluations in the camera and object pose estimation. Also, the object and scene 3D reconstruction should also be evaluated. Since the aim is to recover the full geometry, it is expected also to evaluate the unseen part, not just the depth evaluation for the seen surface.


**Summary Of Recommendation:**

The presented paper shows some new ideas and novel results. However, it requires extra evaluations to prove its innovation and also needs more clear explanation in its method part. This makes me on the borderline (tends to weak accept). The final recommendation will depend on if the mentioned concerns can be solved.

---

> ### Author Response · Authors · 2021-08-31
> **Response to Reviewer jaEn**
>
> We would like to thank reviewer jaEn for the feedback, appreciation of our work’s novelty, and for providing useful suggestions on how to improve the paper. Below we cover the points raised in the feedback.
>
>
> > Related work.
>
> - Thank you for mentioning these additional related works. Both of them were added to the related works section now. We agree that one of the differences between the mentioned methods and ours is our model’s ability to recover the full appearance of the instances as well as occluded background. Moreover, our differentiable object representation in combination with presented optimization procedure allows for further appearance optimization, which significantly improves the reconstruction results as can be seen in the added ablation study - please refer to supplemental Section 3.1.
>
>
> > Additional qualitative and quantitative results.
>
> - As we mention in A1 of our general response, we have generated additional qualitative view synthesis results - please refer to Figure 6 in the updated supplementary. In addition to the original comparison against PixelNerf, we added a second baseline using the Packnet-SfM monocular depth estimation architecture (ref [2] in the updated manuscript) as requested by ky9j. As can be seen from the qualitative results, PixelNeRF trained on monocular sequences fails to generalize well to a stereo view-point providing a highly impaired noisy and blurry result. The PackNet baseline generates output by warping original RGB images using regressed depth maps. This baseline results in sharp images, but significantly distorts foreground objects due to predicted depth inaccuracy. Finally, our method preserves object geometry by using our differentiable renderer, but still retains PackNet-like artifacts on the background due to similar depth-estimation limitations. Additionally, we provide a diverse set of reconstructions demonstrating the performance of our system on a variety of different scenes from different datasets  - please refer to Figures 7-9 in the updated supplementary.
>
> - As we mention in A2 of our general response, we have performed an ablative analysis of different components of our optimization pipeline for full object shape estimation using a bidirectional Chamfer distance, luminance of the recovered objects, and 2D amodal mask alignment comparing the mask for the entire unoccluded vehicle with the respective ground truth. Our results (see Table 1 in the updated supplementary) demonstrate a significant improvement with respect to the selected metrics thanks to optimization and also suggest that both 2D and 3D alignment steps are complementary to each other and provide better results in combination.
>
>
> > The method presented can recover the full appearance of detected instances, however this point is not evaluated.
>
> - We could not compute full appearance metrics in occluded areas as we don’t have access to that ground truth information; instead, we generated additional qualitative reconstructions which capture the appearance of objects and background in areas that are unseen in the original images (please refer to Figures 7-9 in the updated supplementary).
>
>
> > Section 3.2 needs a more clear explanation of the losses used.
>
> - As we mention in A3 in our general response, we have restructured and improved Section 3.2, starting with a high-level introduction and pointing to the relevant subsections for detailed information. We also added additional information on the optimization losses, with full details in the updated supplementary in Section 2.
>
>
> > L150: Fig.5a is not the figure referred to (it does not include the object prior database).
>
> - Thank you for pointing out the typo. It was supposed to point to Fig. 2a. We have now corrected it.

---

> > ### Comment · Reviewer_jaEn · 2021-09-01
> > **Some unaddressed concerns**
> >
> > I would like to thank the authors very much for making efforts to polish the paper and adding extra results. I agree it is an interesting paper, however, my concern has not been well addressed.
> >
> > * Evaluating the recovery of the full appearance:
> > the main advantage over the mentioned papers that can even recover the room layout and generate multi-class instances is it is able to recover the full appearance of the detected objects. Other components, such as utilizing NOCS, learning SDF prior, or optimisation over a learned shape prior is not a very new thing. However, this point is still not shown in the rebuttal. This is the part I am interested to see and really worth some evaluations. Fig.7\~9 only showed the seen parts. I have also checked carefully on the new supplementary video. 01:00~01:11 contains some reasonable appearance recovery, however, the remaining two examples, 01:12-01:23 and 01:26-01:35 showed some corrupted results. Object-wise geometry and appearance recovery examples are also not shown in those examples.
> >
> > 2. Explanation of losses: the authors further the loss function used in the 2D and 3D optimisation. However, the question of each individual loss term is still unaddressed. It further brings more unexplained loss notations in Section 2.
> >
> > 3. Evaluation of object pose estimation: the concern on evaluating the object pose estimation performance is unaddressed. The paper claimed it as one contribution and used PnP and further 2D/3D optimisations to refine the object poses. I am not sure why this part is ignored again.
> >
> > Another small concern. A comparison to PixelNeRF is added to show qualitative view synthesis results. However, it is tested in a dynamic scene and PixelNeRF assumes a static environment. I don't think it is a fair comparison. It is expected that PixelNeRF will fail in this test setting and the blurring caused by moving objects in neural radiance field approaches is well-known. A more fair comparison to the neural radiance field approaches with be to compare with "Neural Scene Graphs for Dynamic Scenes, Julian Ost et al.", which also proposed an object-level view synthesis approach.
> >
> > Despite these unaddressed points, the paper works on a related problem in CoRL and proposed an interesting solution to it. Considering these, I tend to "Weak Reject: I recommend rejecting the paper, but will not argue for my recommendation if the majority of other reviewers have a different opinion.".

---

> > > ### Author Response · Authors · 2021-09-03
> > > **Response to Reviewer jaEn (Part 1/2)**
> > >
> > > Thank you very much for your response. We’d like to answer your remaining questions, and are happy to continue discussing.
> > >
> > > > Evaluating the recovery of the full appearance
> > >
> > > We agree that evaluating the full appearance of the objects regressed by our method would be an interesting additional experiment. However, as mentioned in our rebuttal, our data does not contain ground truth for unseen object parts, making it impossible for us to conduct this experiment. Instead, we have opted for qualitative evaluations. We’d like to further point out that Figures 7 - 9 in the supplementary do indeed show object parts that were not present in the input images. Specifically, in Figure 7, the bottom left subfigure shows the white car from a completely new angle: the left side of the white car is not visible at all in the input RGB image. Similarly, in the bottom middle subfigure of Figure 7, the right side of the gray car is not visible in the input image, but is displayed in our model’s reconstruction. We argue that these examples show that our method is capable of recovering both the object’s geometry and appearance, and that they are rendered with high fidelity under novel viewpoints. We further provide a video of the reconstructed pointcloud scenes: https://streamable.com/964grj (note that we show both the input images as well as the resulting reconstructions).
> > >
> > >
> > > >Missing evaluation of object pose estimation
> > >
> > > Thank you for bringing this to our attention. We would like to present an additional experiment that quantifies the effect of our optimization procedure; we will update the final paper to include these results.
> > >
> > > We note that our initial pose, as regressed by an outlier-robust PnP solver, is usually already very accurate for a given NOCS prediction. However, since we optimize both pose and shape at the same time, allowing pose to change ensures the maximal alignment. To confirm this we ran an additional ablation for four 2D optimization modalities - no optimization, only shape, only pose, both pose and shape. We quantify the pose regressed using (i) the median angular error comparing the ground truth pose $\textbf{q}$ and the optimized pose $\hat{\textbf{q}}$; and (ii) for translation, we estimate the difference between the ground truth translation and the optimized one considering only X and Y components. Additionally, we show the performance of shape estimation using the median bidirectional Chamfer distance to see how different modalities perform in this context. We performed this experiment on a subset of the PD test set containing 200 images, with each image belonging to a different scene: $\theta(\textbf{q}_i, \hat{\textbf{q}_i}) = 2 \cdot arccos(|\textbf{q}_i \cdot \hat{\textbf{q}_i}|)$.
> > >
> > > |                 | No opt. | Shape | Pose  | Shape + Pose |
> > > |-----------------|---------|-------|-------|--------------|
> > > | Angle (deg)     | 5.99    | 5.99  | 6.55  | 5.97         |
> > > | Chamfer (mm)    | 14.72   | 9.90  | 14.72 | 8.80         |
> > > | Translation (m) | 4.00    | 4.00  | 3.98  | 3.99         |
> > >
> > > From the results we can see that optimizing the pose alone results in pose degradation due to ignoring the shape component. However, optimizing both shape and pose at the same time ensures that pose performance does not degrade. We note a similar result for the shape benchmark - while still improving results when only the shape component is optimized, combining both shape and pose results in the best overall performance. As we can also see the translational component stays largely unchanged.
> > >
> > > As shown in our supplementary Table 1 - we have also measured the effect of different optimization components on object shape reconstruction, object luminance as well as amodal mask alignment. Moreover, Table 3 of the main paper demonstrates the 3D detection performance of our pipeline, which is a good proxy for pose performance.
> > > Additionally, we include a video demonstrating the process of 2D optimization with a displaced initial pose: one can see that despite a not very precisely predicted NOCS pose we are able to adapt the pose and shape to achieve a maximum possible alignment: https://streamable.com/l6k0q4

---

> > > ### Author Response · Authors · 2021-09-03
> > > **Response to Reviewer jaEn (Part 2/2)**
> > >
> > > > Explanation of losses
> > >
> > > In Section 2 of the supplementary material we provide separate formulations for both 2D and 3D loss types. Both loss types are defined as simple norms and we therefore only provided generalized formulations. We would like to explain terms here and we will further refine Section 2 explaining each term separately.
> > >
> > > - 2D losses
> > >   - The 2D NOCS loss compares the predicted NOCS map coming from the NOCS head of the 3D reasoning block with the NOCS output of the renderer. Moreover, we limit the loss to the subset pixels contained in the union of predicted and rendered masks, denoted by $P$.
> > > $$L_{NOCS_{2D}} =  \frac{1}{|P|}\sum_{p \in P} || nocs^{pred}_p -  nocs^{ren}_p ||_2$$
> > >   - The 2D Luminance Field loss compares the input RGB image with the RGB output of the renderer
> > > $$L_{LF_{2D}} =  \frac{1}{|P|}\sum_{p \in P} || lf^{pred}_p -  lf^{ren}_p ||_2$$ Similarly, we only consider pixels contained in the union of predicted and rendered masks.
> > >   - The 2D mask loss compares the predicted mask coming from 3D reasoning block with the mask output of the renderer for the pixels contained in the union of both masks
> > > $$L_{mask_{2D}} =  \frac{1}{|P|}\sum_{p \in P} || mask^{pred}_p -  mask^{ren}_p ||_2$$
> > > - 3D losses
> > >   - For each detected instance we recover its partial shape using predicted NOCS map resulting in points  $\mathbf{p} = {p_1, ..., p_n}$. Additionally, given predicted latent codes $z_{sdf}$ and $z_{lf}$ we recover a full colored prior shape in the form of a colored SDF field, which we transform to a colored point cloud using 0-isosurface projection $\mathbf{c} = \{c_1, ..., c_k\}$. Then for each point $p_i$ we determine the nearest neighbor from $c$ and keep it if it is closer than 0.2m. Finally, the loss is calculated as the mean distance over all correspondences $C_{3D}$:
> > > $$L_{SDF} =  \frac{1}{|C_{3D}|} \sum_{(i,j) \in C_{3D}} || p_i - c_j ||_1$$
> > >   - Our 3D Luminance Field loss is defined similarly to the SDF loss, but instead of point coordinates, we compare RGB colors between the partial reconstruction based on predicted NOCS and using input RGB colors and our recovered colored object point cloud:
> > > $$L_{LF} =  \frac{1}{|C_{3D}|} \sum_{(i,j) \in C_{3D}} || p^{rgb}_i - c^{rgb}_j ||_1$$
> > >   - Finally, our symmetry loss works by minimizing the difference between the left and right sides of our reconstructed colored object point cloud. For that we first compute nearest neighbors between the left side and an inverted right side of the object and then minimize the L1 distance between these correspondences.
> > > $$L_{symm} =  \frac{1}{|C_{3D}|} \sum_{(i,j) \in C_{3D}} || l_i - r_j ||_1.$$
> > >
> > > > Comparison to PixelNeRF
> > >
> > > We agree that PixelNeRF has its limitations and assumes a static environment. However, we note that our setting is that of 3D reconstruction from a single image. Dynamic NeRF methods as mentioned by the reviewer require that the object has been observed from every perspective, and are therefore not applicable to this setting. PixelNeRF is the state-of-the-art single-image 3D reconstruction method, and the method that assumes a setup closest to ours.
> > > The method “Neural Scene Graphs for Dynamic Scenes” [Ost et al.] is similarly not applicable to our setting, as it requires 3D bounding boxes and object class for each object in the scene, while we operate on single images.

---

### Official Review · Reviewer_JgCa · 2021-07-21

**Originality:** Good
**Technical Quality:** Good
**Clarity Of Presentation:** Very Good
**Impact:** 4

**Recommendation:**

Weak Accept: I recommend accepting the paper, but will not argue for my recommendation if the majority of other reviewers have a different opinion.

**Summary:**

This paper proposes a method for reconstructing, and decomposing a scene from a single RGB image. The main idea is to use a decomposition network to simultaneously predict normalized object coordinates a depth map of the background. The object poses can then be easily computed using PnP and refined using a database of object models and an optimization procedure. The results show that the proposed approach generalizes well and achieves excellent performance on view synthesis, depth estimation and object detection in this domain.


**Issues:**

I would like the authors to address the following in the rebuttal (or final revision):
- How accurate are the different stages of the pipeline? I.e. how does the accuracy of the PnP compare to the final optimized result?
- How is the scale factor determined at testing time, is there a fixed scale for each of the 27 database objects?
- It would be good to provide more qualitative examples to show the performance of the approach across different scenes. When does the method fail? And when does it work really well?
- How would the approach work with vehicles of other shapes like buses, vans or bicycles?


**Reviewer Expertise:**

Good: General knowledge of the area

**Strengths And Weaknesses:**

**Strengths**
- The paper is very well written from start to finish. It is clear that the authors have spent a lot of effort on writing and polishing the paper. The paper is easy to follow and describes the method in sufficient detail (there is, however, some information that is missing related to the data used, see below).
- The setup of the problem is very interesting and is the way I think perception problems should be addressed. By this I mean instead of just doing black-box depth prediction / object detection, the scene is decomposed in a more “symbolic” manner and the combination of each of these elements helps to describe the overall scene. This allows the method to be used for multiple tasks at the same time in an interpretable manner which is not often the case with deep learning approaches.
- The evaluation is targeted and supports the claims made in the paper. In particular cross-domain depth estimation is tested, 3D object detection accuracy is tested and view synthesis is tested.
- The results are good both quantitatively and qualitatively. The supplementary video showing the fully reconstructed scene and cars is quite impressive (however, it would be nice to see more examples).
- The method shows great generalization capability. It is entirely trained on synthetic images (Parallel domain and VKITTI2) and is able to generalize well to real KITTI and DDAD.

**Weaknesses**
- The approach works only for a single class of object. For each new class, a database of object models is required that have been placed in a canonical frame of reference. This is ok for objects that don’t vary much in shape or appearance (like cars) but won’t work well for general objects. Of course, in the scenario autonomous / assisted driving this is not too much of an issue. However, it does limit the application to other areas.
- There are not many example scenes shown in the results. There are only two individual frames in the supplementary video and one in the main paper. Since the paper addresses the problem of view synthesis it would be good to see a more diverse range of reconstructions to get a better idea of the quality.
- There is not indication of the runtime of the method and how stable it is when applied to video prediction (which is how it would probably be applied in the autonomous vehicle / ADAS case).


**Summary Of Recommendation:**

This is a good paper in the sense that it presents an interesting method and achieves some solid results. However, the weaknesses I’ve considered in the recommendation are the fact that it will be difficult to adapt to other setups (apart from road scenes), and the paper limited examples showing how the approach works.

---

> ### Author Response · Authors · 2021-08-31
> **Response to Reviewer JgCa**
>
> We would like to thank reviewer JgCa for the positive feedback, appreciation of our work’s presentation, methodology, and results, as well as useful suggestions. Below we answer the questions raised in the feedback.
>
> > Additional qualitative results + failure cases.
>
> - As we mention in A1 of our general response, we have generated additional qualitative view synthesis results - please refer to Figure 6 in the updated supplementary. In addition to the original comparison against PixelNerf, we added a second baseline using the Packnet-SfM monocular depth estimation architecture (ref [2] in the updated manuscript) as requested by ky9j. As can be seen from the qualitative results, PixelNeRF trained on monocular sequences fails to generalize well to a stereo view-point providing a highly impaired noisy and blurry result. The PackNet baseline generates output by warping original RGB images using regressed depth maps. This baseline results in sharp images, but significantly distorts foreground objects due to predicted depth inaccuracy. Finally, our method preserves object geometry by using our differentiable renderer, but still retains PackNet-like artifacts on the background due to similar depth-estimation limitations. Additionally, we provide a diverse set of reconstructions demonstrating the performance of our system on a variety of different scenes from different datasets  - please refer to Figures 7-9 in the updated supplementary. Moreover, supplemental Section 3.4 also provides a discussion on the failure modes and limitations of our pipeline.
>
> > How would the approach work with vehicles of other shapes like buses, vans or bicycles? The approach seems to be limited to rigid objects; how would this work for general objects?
>
> - From our preliminary experiments, extending our pipeline to work with other vehicles, such as buses and vans, functions similarly well. It is however less trivial for classes representing smaller objects, like bicycles, and non-rigid objects, like pedestrians. Smaller affected pixel area results in a less robust detection as well as a coarser and lower quality shape reconstruction. Yet another problem arises from the current implementation of PriorDB that stores only rigid objects and would require additional effort to extend it to work with partially and fully non-rigid objects. We added the discussion to the supplemental Section 3.4 and plan to address the extension to other classes and underlying challenges in future work.
>
> > Runtime analysis.
>
> - As mentioned in A2 of our general response, we report the following runtime statistics evaluated on a Titan Xp GPU. A partial scene reconstruction takes 0.9s on average, but could be optimized to reach real-time performance. Object optimization on the other hand takes around 18s per object for 50 optimization steps and is not real-time capable at the moment. We will take an effort to optimize the performance for both steps in future work.
>
>
> > How is the scaling factor determined at test time?
>
> - Our PriorDB retains relative scales by using a single scale factor for all objects within it. Therefore, we rely on the scales presented in PD and use them as an additional prior. Alternatively, one could regress the scale factor directly from the DETR features.
>
>
> > How stable is the method when applied to video prediction?
>
> - Our method operates on single frames, and we focused on evaluating various tasks within that context. We agree that extending to multi-frame is interesting future work.

---

### Official Review · Reviewer_pv1C · 2021-07-22

**Originality:** Good
**Technical Quality:** Good
**Clarity Of Presentation:** Poor
**Impact:** 2

**Recommendation:**

Weak Accept: I recommend accepting the paper, but will not argue for my recommendation if the majority of other reviewers have a different opinion.

**Summary:**

This paper proposes a holistic single-shot scene reconstruction system that separately recovers the depth of the background and the 3D geometry and appearance of the objects.

**Issues:**

As stated in weakness.

**Reviewer Expertise:**

Fair: Some knowledge of the area

**Strengths And Weaknesses:**

Strength:
- Results are strong.
- It's an interesting idea to use SDF features to encode the prior dataset and fit the reconstructed partial object with the learned SDF decoder.

Weakness:
- The structure of section 3.2 is very messy:
  - You have to read 3.2.1 and 3.2.2 before you can understand the first part of 3.2. Because this part is not high-level enough and many unexplained terms appear in this part.
  Please briefly introduce the terms like PriorDB feature, 0-isosurface projection, and Lsymm loss.
  - 158: The distance between which point clouds? Is the ground truth full point cloud of the objects available? Or partial point clouds are compared?
- 207: Fig 5 does not have an "a
- It makes sense to fit SDF of prior shapes because cars' shapes are close. But the luminance field can be very different across the shapes. For example, in Figure 5, the rendered color of the second car on the right side of the road is orange, while in partial reconstruction, it's white.
- No ablation study is conducted. For example, w/ or w/o the 3 components in 2D alignment.
- Few qualitative results are shown in paper or supplementary material. No qualitative results of novel view synthesis of PixelNeRF and the proposed methods are shown, which makes the discussion not convincing.

**Summary Of Recommendation:**

The idea is pretty cool. But the lack of qualitative results and the poor writing harms the quality of the paper. So I recommend rejecting the paper. But if the author can give a concrete rebuttal, I'll definitely reconsider.

Update: The authors' responses address my questions. So I would like to change my recommendation to 'weak accept'.

---

> ### Author Response · Authors · 2021-08-31
> **Response to Reviewer pv1C**
>
> We would like to thank reviewer pv1C for the feedback, appreciation of our method’s novelty and strong results as well as for the useful suggestions on how to improve our work. Below we answer the questions raised in the feedback.
>
>
> >Section 3.2.
>
> - As we mentioned in A3 in our general response, we have restructured and improved Section 3.2, starting with a high-level introduction and pointing to the relevant subsections for detailed information. We also added additional information on the optimization losses, with full details in the updated supplementary in Section 2.
>
>
> > Ablation study.
>
> - As we mentioned in A2 of our general response, we have performed an ablative analysis of different components of our optimization pipeline for full object shape estimation using a bidirectional Chamfer distance, luminance of the recovered objects, and 2D amodal mask alignment comparing the mask for the entire unoccluded vehicle with the respective ground truth. Our results (see Table 1 in the updated supplementary) demonstrate a significant improvement with respect to the selected metrics thanks to optimization and also suggest that both 2D and 3D alignment steps are complementary to each other and provide better results in combination.
>
> > Additional qualitative results.
>
> - As we mention in A1 of our general response, we have generated additional qualitative view synthesis results - please refer to Figure 6 in the updated supplementary. In addition to the original comparison against PixelNerf, we added a second baseline using the Packnet-SfM monocular depth estimation architecture (ref [2] in the updated manuscript) as requested by ky9j. As can be seen from the qualitative results, PixelNeRF trained on monocular sequences fails to generalize well to a stereo view-point providing a highly impaired noisy and blurry result. The PackNet baseline generates output by warping full-resolution original RGB images using regressed depth maps. This baseline results in sharp images, but significantly distorts foreground objects due to predicted depth inaccuracy. Finally, our method preserves object geometry by using our differentiable renderer, but still retains PackNet-like artifacts on the background due to similar depth-estimation limitations. Additionally, we provide a diverse set of reconstructions demonstrating the performance of our system on a variety of different scenes from different datasets  - please refer to Figures 7-9 in the updated supplementary.
>
>
> > It makes sense to fit SDFs of prior shapes because cars' shapes are close. But the luminance field can be very different across the shapes.
>
> - We agree that fitting luminance fields is more difficult than fitting SDF, that’s why we allow the LF module weights to vary during the optimization. Alternatively, we would be limited to the luminance distribution presented in the used synthetic dataset (Parallel Domain). We also apologize for the confusion caused by Figure 5 due to the 2 images coming from 2 different scenes. We have corrected it in the updated version.
>
>
> > line 158: The distance between which point clouds?
>
> - We refer to the partial point cloud recovered from predicted NOCS maps and a full shape point cloud predicted by PriorDB. An example of this optimization is shown in Figure 1 (NOCS <-> SDF/LF) and in the supplementary video.
>
>
> > 207: Fig 5 does not have an "a”.
>
> - Thank you for pointing out the typo. It was supposed to point to Fig. 2a. We have now corrected it.

---

> ### Author Response · Authors · 2021-09-04
> **Response to Reviewer pv1C**
>
> Dear Reviewer pv1C,
>
> We hope we were able to address your concerns in our rebuttal. Please let us know if you would like to reconsider your rating, or if you have any remaining questions - we'd be happy to answer them and continue discussing. Thank you!

---

### Official Review · Reviewer_ky9j · 2021-08-17

**Originality:** Very Good
**Technical Quality:** Very Good
**Clarity Of Presentation:** Good
**Impact:** 4

**Recommendation:**

Strong Accept: I recommend accepting the paper and will argue for my recommendation even if other reviewers hold a different opinion.

**Summary:**

This paper introduces a complete pipeline to represent a scene with 3D model of moving objects and immovable background by a single image. The approach utilize NOCS-SDF, RGB-LF and surfel-based differentiable rendering to optimize the full geometry of movable object instances with the help of a object prior database. The decomposition of the scene enables the approach to edit the movable objects and re-render the scene.

**Issues:**

- Will the transformer-based feature extractor harm the performance of the depth estimation module? We have not seen many literature addressing this issue, and you are actually sharing the representation with two tasks: Depth and NOCS. A ablation study which shows the advantage of transformer-based feature extractor will be nice to have.
- Why this approach can apply to real data without domain adaptation? Do you mean by similar scene like the example you showed in the paper: VKITTI and KITTI? If the scene changes, will it still be able to have a strong sim-to-real performance without domain adaptation?
- The missing loss functions in the Section 3.2 can make the reader without related background knowledge hard to understand.
- Can we have a way to show the advantage of partial shape recevery by quantitative analysis? This seems a crucial step to obtain a good final object reconstruction with differential render.

**Reviewer Expertise:**

Very good: Comprehensive knowledge of the area

**Strengths And Weaknesses:**

Strengths
- The idea decomposing the scene to dynamic objects and static scene is quite commonly exploited. However, the representation of the object this paper developed has a good geometry representation. It can also recover the surface of object based on a prior object database. The combination of SDF and LF MLP makes sense from the perspective of theory and shows promising result.
- This paper introduces the surfel-based differentiable renderer into this field for the first time. Comparing with NeRF, the surfel-based differentiable renderer is more geometrical consistent. It will not collapse when you generalize this approach to unseen scene.

Weakness
- The effectivenes of transformer-based feature extractor needs to be validated. Usually transformer-based architecture requires more data to keep the same performance as CNN. Is the transformer-based feature extractor really better than CNN in this task? Also, it would be misleading to use CNN instead of Transformer in the Figure 1.
- It is not fair to compare this method with NeRF since NeRF deals with novel view synthesis. The fundamental principle of NeRF and this approach have a large discrepancy. It would be fair to compare it with monocular depth estimation with editable moving objects, which shares the same representation of this approach. You can compare different object representation under the framework of monocular depth estimation.
- The details of object reasoning block are missing in the paper and supplementary material.
- The authors claim this approach can apply to real data without domain adaptation. It would be nice if a quantitative result of the performance difference between VKITTI dataset and KITTI dataset.

**Summary Of Recommendation:**

This work solve the monocular scene reconstruction by scene decomposition of dynamic objects and static scene. By introducing NOCS-SDF and RGB-LF based optimization and surfel-based differentiable render, the novel representation and rendering of these movable instances makes this approach generalizable to unknown scene (but not for unknown objects). The theory of this approach is solid and the overall design is reasonable for robotics application.

---

> ### Author Response · Authors · 2021-08-31
> **Response to Reviewer ky9j**
>
> We would like to thank reviewer ky9j for the feedback and useful suggestions. We also appreciate the acknowledgment of our work’s object representation and generalizability to unknown scenes. Below we answer the questions raised in the feedback.
> > Quantitative evaluation showing the performance difference between the VKITTI dataset and the KITTI dataset.
> - Thank you for the suggestion. We have included depth estimation results evaluating in-domain synthetic performance (VKITTI to VKITTI and PD to PD), using our proposed pipeline as well as the two others considered in the paper (Monodepth2 and PackNet). These results suggest that our method does not just transfer between synthetic and real-world domains (as demonstrated in the paper, Table 1), but also enables a better in-domain performance for foreground objects. We attribute this behavior to the fact that depth networks operate at a per-pixel level, minimizing an objective that takes into consideration the entire image. This includes mostly background structures (e.g., road and buildings), making the depth network learn to accurately predict these structures at the expense of foreground objects. Please refer to Table 2 in the updated supplementary for more details.
>
> > Quantitative analysis of advantages given by partial shape recovery.
> - We added an ablation study demonstrating the importance of partial shape recovery for full shape optimization - please see supplemental Section 3.1. As can be seen from the results, the partial shape estimation is crucial to robustly recover the full shape geometry and it significantly improves the final shape estimate. Moreover, it is also used to estimate the object’s pose by maximizing its alignment in the projective space, which is essential to be able to place and render scene objects.
>
> > Comparison against a monocular-depth estimation baseline.
> - As we mentioned in A1 and A2 of our general response, we have added Packnet-SfM as an additional baseline, with qualitative results in Figure 6 in the updated supplementary, and quantitative view synthesis results in the updated Table 1 in the main submission. As can be seen from the qualitative results, PixelNeRF trained on monocular sequences fails to generalize well to a stereo view-point providing a highly impaired noisy and blurry result. The PackNet baseline generates output by warping original RGB images using regressed depth maps. This baseline results in sharp images, but significantly distorts foreground objects due to predicted depth inaccuracy. Finally, our method preserves object geometry by using our differentiable renderer, but still retains PackNet-like artifacts on the background due to similar depth-estimation limitations. Quantitatively, our method outperforms both PackNet-SfM and PixelNerf for view synthesis, as can be seen from the updated Table 2.
>
> > What makes the approach generalize well to real data without domain adaptation? Will it still perform well in less similar sim and real datasets?
> - NOCS generalizes well to unseen real data due to an integrated object prior and discretized solution space; we added additional details describing this in Section 1.2 of the supplementary material. We formulate NOCS regression as a classification problem, which significantly decreases the size of the solution space (discrete $256^3$ for classification and infinite continuous solution space $[-1; 1]^3$ in the case of direct regression). This successful transfer is well shown in our depth and 3D detection experiments. Background regression however is more prone to overfitting and shares generalization abilities similar to most CNN-based depth estimation pipelines.
>
> > Transformer vs CNN based architectures for feature extraction:
> - We chose the DETR transformer architecture owing to its simplicity (no hand-tuning of anchors or non-maxima suppression) as well as its compelling state-of-the-art performance for object detection. Our choice of architecture is further validated by the strong results we obtained for the tasks considered. We agree that a CNN-based architecture could be used here, but we view that as an orthogonal experiment and leave it as future work.
>
> > The details of object reasoning block.
> - We have added more technical details to further explain the object reasoning block (Section 3.1 of the main paper).
>
> > Surfel-based differentiable renderer.
> - As mentioned in the paper, our surfel-based renderer is based on [54 in updated paper], which we extend to improve its efficiency, making it suitable for our work. We have further clarified this in the text - please see supplemental Section 1.4.
>
> > Loss information in Section 3.2.
> - As we mentioned in A3 in our general response, we have restructured and improved Section 3.2, starting with a high-level introduction and pointing to the relevant subsections for detailed information. We also added additional information on the optimization losses, with full details in the updated supplementary in Section 2.

---

### Author Response · Authors · 2021-09-01
**Response to Meta Reviewer**

(Copy of the original response to Meta Reviewer posted for public visibility)

First of all, we would like to thank all the reviewers and the AC for their feedback and useful suggestions. We are glad to see the recognition of the novelty of our work (ky9j, pv1C, jaEn), relevance to the field (JgCa), clarity of presentation (JgCa), strong results (pv1C) on view synthesis, depth estimation and object detection (JgCa), and generalizability to unknown scenes (ky9j, JgCa).

Below, we respond to the most important questions raised by the reviewers, and we provide individual responses to each reviewer addressing any remaining questions.

> Q1. [All] Additional qualitative results.

A1:

- We have generated additional qualitative view synthesis results - please refer to Figure 6 in the updated supplementary. In addition to the original comparison against PixelNerf, we added a second baseline using the Packnet-SfM monocular depth estimation architecture (ref [2] in the updated manuscript) as requested by ky9j. As can be seen from the qualitative results, PixelNeRF trained on monocular sequences fails to generalize well to a stereo view-point providing a highly impaired noisy and blurry result. The PackNet baseline generates output by warping original RGB images using regressed depth maps. This baseline results in sharp images, but significantly distorts foreground objects due to predicted depth inaccuracy. Finally, our method preserves object geometry by using our differentiable renderer, but still retains PackNet-like artifacts on the background due to similar depth-estimation limitations.
- As requested by reviewers, we provide a diverse set of reconstructions demonstrating the performance of our system on a variety of different scenes from different datasets - please refer to Figures 7-9 in the updated supplementary.

> Q2. [pv1C, JgCa, jaEn, ky9j] Improved evaluation and runtime analysis.

A2:

- Ablative analysis: As requested, we have performed an ablative analysis of different components of our optimization pipeline for full object shape estimation using a bidirectional Chamfer distance, luminance of the recovered objects, and 2D amodal mask alignment comparing the mask for the entire unoccluded vehicle with the respective ground truth. Our results (see Table 1 in the updated supplementary) demonstrate a significant improvement with respect to the selected metrics thanks to optimization and also suggest that both 2D and 3D alignment steps are complementary to each other and provide better results in combination.
- Runtime: We report the following runtime statistics evaluated on a Titan Xp GPU. A partial scene reconstruction takes 0.9s on average, but could be optimized to reach real-time performance. Object optimization on the other hand takes around 18s per object for 50 optimization steps and is not real-time capable at the moment. We will take an effort to optimize the performance for both steps in future work.
- View synthesis: As mentioned, we generated qualitative results using PackNet-SfM in Figure 6 in the updated supplementary. Additionally, we also evaluated PackNet-SfM quantitatively, with results reported in Table 2 in the updated submission. Moreover, we improved the view-synthesis results of our pipeline by additional tuning of rendering parameters. As can be seen from the updated Table 2, our method outperforms both PackNet-SfM and PixelNerf for view synthesis.
- Failure modes: Supplemental Section 3.4 provides a discussion on the failure modes and limitations of our pipeline.

> Q3: [pv1C,jaEn,ky9j] Improved Section 3.2.

A3:

- We have restructured and improved Section 3.2, starting with a high-level introduction and pointing to the relevant subsections for detailed information. We also added additional information on the optimization losses, with full details in the updated supplementary in Section 2.

We will provide answers to the remaining points for each reviewer individually below. Please let us know if any other questions arise - we are happy to discuss further and answer any remaining points. Once again, we sincerely thank all reviewers for their feedback and suggestions, and for their appreciation of our work!

---

### Meta-Review · Area_Chair_SwfN · 2021-08-13

**Recommendation:** Accept (Poster)
**Confidence:** 5

**Metareview:**

All reviewers received this paper positively. Both presentation and results were reported as convincing, however, the ratings were borderline (1x weak reject, 2x weak accept). The main criticism relates to the lack of qualitative results (all reviewers), evaluation shortcomings (pv1C: ablation study, JgCa: runtime and stability analysis, jaEn: accuracy of camera and pose estimation) and presentation (pv1C+jaEn: section 3.2). Given the authors revise the manuscript accordingly, I’d be happy to accept the paper.

---

> ### Author Response · Authors · 2021-08-30
> **Response to Meta Reviewer**
>
> First of all, we would like to thank all the reviewers and the AC for their feedback and useful suggestions. We are glad to see the recognition of the novelty of our work (ky9j, pv1C, jaEn), relevance to the field (JgCa), clarity of presentation (JgCa), strong results (pv1C) on view synthesis, depth estimation and object detection (JgCa), and generalizability to unknown scenes (ky9j, JgCa).
>
> Below, we respond to the most important questions raised by the reviewers, and we provide individual responses to each reviewer addressing any remaining questions.
>
>
> > Q1. [All] Additional qualitative results.
>
> A1:
> - We have generated additional qualitative view synthesis results - please refer to Figure 6 in the updated supplementary. In addition to the original comparison against PixelNerf, we added a second baseline using the Packnet-SfM monocular depth estimation architecture (ref [2] in the updated manuscript) as requested by ky9j. As can be seen from the qualitative results, PixelNeRF trained on monocular sequences fails to generalize well to a stereo view-point providing a highly impaired noisy and blurry result. The PackNet baseline generates output by warping original RGB images using regressed depth maps. This baseline results in sharp images, but significantly distorts foreground objects due to predicted depth inaccuracy. Finally, our method preserves object geometry by using our differentiable renderer, but still retains PackNet-like artifacts on the background due to similar depth-estimation limitations.
> - As requested by reviewers, we provide a diverse set of reconstructions demonstrating the performance of our system on a variety of different scenes from different datasets  - please refer to Figures 7-9 in the updated supplementary.
>
>
> > Q2. [pv1C, JgCa, jaEn, ky9j] Improved evaluation and runtime analysis.
>
> A2:
> - Ablative analysis: As requested, we have performed an ablative analysis of different components of our optimization pipeline for full object shape estimation using a bidirectional Chamfer distance, luminance of the recovered objects, and 2D amodal mask alignment comparing the mask for the entire unoccluded vehicle with the respective ground truth. Our results (see Table 1 in the updated supplementary) demonstrate a significant improvement with respect to the selected metrics thanks to optimization and also suggest that both 2D and 3D alignment steps are complementary to each other and provide better results in combination.
> - Runtime: We report the following runtime statistics evaluated on a Titan Xp GPU. A partial scene reconstruction takes 0.9s on average, but could be optimized to reach real-time performance. Object optimization on the other hand takes around 18s per object for 50 optimization steps and is not real-time capable at the moment. We will take an effort to optimize the performance for both steps in future work.
> - View synthesis: As mentioned, we generated qualitative results using PackNet-SfM in Figure 6 in the updated supplementary. Additionally, we also evaluated PackNet-SfM quantitatively, with results reported in Table 2 in the updated submission. Moreover, we improved the view-synthesis results of our pipeline by additional tuning of rendering parameters. As can be seen from the updated Table 2, our method outperforms both PackNet-SfM and PixelNerf for view synthesis.
> - Failure modes: Supplemental Section 3.4 provides a discussion on the failure modes and limitations of our pipeline.
>
> > Q3: [pv1C,jaEn,ky9j] Improved Section 3.2.
>
> A3:
> - We have restructured and improved Section 3.2, starting with a high-level introduction and pointing to the relevant subsections for detailed information. We also added additional information on the optimization losses, with full details in the updated supplementary in Section 2.
>
>
>
> We will provide answers to the remaining points for each reviewer individually below. Please let us know if any other questions arise - we are happy to discuss further and answer any remaining points. Once again, we sincerely thank all reviewers for their feedback and suggestions, and for their appreciation of our work!

---

### Decision · Program_Chairs · 2021-09-13

**Decision:**

Accept (Poster)

**Comment:**

All reviewers received this paper positively. Both presentation and results were reported as convincing, however, the ratings were borderline (1x weak reject, 2x weak accept). The main criticism relates to the lack of qualitative results (all reviewers), evaluation shortcomings (pv1C: ablation study, JgCa: runtime and stability analysis, jaEn: accuracy of camera and pose estimation) and presentation (pv1C+jaEn: section 3.2). Given the authors revise the manuscript accordingly, I’d be happy to accept the paper.